# Particulate Matter in an Urban–Industrial Environment: Comparing Data of Dispersion Modeling with Tree Leaves Deposition

**Gregorio Sgrigna** [1,*] **, Hélder Relvas** [2] **, Ana Isabel Miranda** [2] **and Carlo Calfapietra** [1]

1 Institute of Research on Terrestrial Ecosystems (IRET), National Research Council (CNR), 05010 Porano (TR), Italy; carlo.calfapietra@cnr.it

2 Centre for Environmental and Marine Studies (CESAM), Department of Environment and Planning, University of Aveiro, 3810-193 Aveiro, Portugal; helder.relvas@ua.pt (H.R.); miranda@ua.pt (A.I.M.)

* Correspondence: gregorio.sgrigna@iret.cnr.it

**Abstract:** Particulate matter represents a serious hazard to human health, and air quality models contribute to the understanding of its dispersion. This study describes particulate matter with a $\leq 10\ \mu m$ diameter (PM10) dynamics in an urban–industrial area, through the comparison of three datasets: modeled (TAPM—The Air Pollution Model), measured concentration (environmental control stations—ECS), and leaf deposition values. Results showed a good agreement between ECS and TAPM data. A steel plant area was used as a PM10 emissions reference source, in relation to the four sampling areas, and a distance/wind-based factor was introduced (Steel Factor, SF). Through SF, the three datasets were compared. The SF was able to describe the PM10 dispersion values for ECS and leaf deposition ($r^2 = 0.61$–0.94 for ECS; $r^2 = 0.45$–0.70 for leaf); no relationship was found for TAPM results. Differences between measured and modeled data can be due to discrepancies in one district and explained by a lack of PM10 inventory for the steel plant emissions. The study suggests the use of TAPM as a suitable tool for PM10 modeling at the urban scale. Moreover, tree leaves are a low-cost tool to evaluate the urban environmental quality, by providing information on whether and when data from leaf deposition can be used as a proxy for air pollution concentration. Further studies to include the re-suspension of particles as a PM10 source within emission inventories are suggested.

**Keywords:** urban air quality; dispersion modeling; air pollution monitoring; urban trees; deposition sampling; particulate matter

## 1. Introduction

The large changes in anthropogenic emissions both within and outside Europe, especially beginning from the first years of the 1990s, has led to a considerable reduction of air pollution. Despite this, relevant concentrations of PM10 (particulate matter with a $\leq 10\ \mu m$ aerodynamic diameter), ozone ($O_3$), and nitrogen dioxide ($NO_2$) are still an important hazard for human health in urban environments [1]. The effects of air pollution are mainly felt in urban areas, where more than half of the world population lives, with an increasing trend of urbanization. By 2050, it is estimated that more than twice as many people in the world will be living in urban (6.7 billion) than in rural settings (3.1 billion). Epidemiological studies in Europe, in Asia, or North America have evidenced the associated high-mortality factors for ambient air pollution, which is a relevant threat for human health in cities [2–6]. In western, central, and eastern Europe, over 400,000 premature deaths could be attributed to exposure to ambient particulate matter [7]. A clear indication to enhance environmental and human conditions has been given by the United Nations by drawing up the 17 Sustainable Development Goals to be adopted before 2030 [8]. An important number of these ambitious goals involve, or are relevant to, urban environments and air pollution.

The European Commission, through the Directive on Ambient Air Quality and Cleaner Air for Europe (Directive 2008/50/EC), stated the importance of protecting human health and the environment as a whole and combating the emissions of air pollutants. Despite the strategies aimed at mitigating the risks deriving from human exposure to different kinds of pollutants, there still are several European areas with concentration levels exceeding the Directive limit values and it is crucial to go further. A concrete approach is represented by the implementation of Nature-based Solutions (NBS) in urban environments [9–11]. Among other benefits, urban trees provide Ecosystem Services (ES) and can be used as bioindicators for urban environmental quality. Furthermore, the European Commission has been highlighting the need for valuing and promoting urban forests and urban trees.

The successful implementation of air quality management strategies and measures to mitigate air pollution depends on the strength of their main components, e.g., goal/objective, emission inventory, monitoring network, air quality modeling, control strategies, and public involvement [12]. Air quality models can reproduce and help one to understand the physical and chemical transformations and the removal processes of gaseous and particulate pollutants in the atmosphere. They contribute to the understanding of complex dynamics and the relationship between urban trees and air pollution, leading to the optimization of ecosystem services provided by NBS in urban areas [13].

The general aim of this study was to describe the PM10 dispersion and deposition in an urban–industrial area by evaluating the relationships between PM10 air concentration data and PM10 urban tree leaf deposition data. The investigated area was the city of Terni (Central Italy), which is well known as one of the most polluted cities in the region. First, an air quality model (The Air Pollution Model—TAPM) was applied to the study area for one year and results were compared with data from environmental control stations (ECS). Secondly, we evaluated the relationships between PM10 air concentration and PM10 leaf deposition data, and finally, the impact of the main fixed air pollutants' source in the city was assessed, based on PM10 concentration and deposition data and using a PM10 urban tree retention place-dependent analysis. Thus, information about PM10 concentration and leaf deposition seasonal/spatial variability was provided, and a model approach at the urban scale was tested. Furthermore, through a newly introduced wind and distance-based spatial factor, the PM10 dispersion and deposition differences among four districts were evaluated.

The study investigated the following four hypotheses:

- There is a seasonal variability of PM10 dispersion in the study area across the year that can be depicted through model predictions, ECS data, and tree leaves deposited PM10;
- Model predictions are comparable with PM10 data from ECS in time (1 year) and space (four districts in the city);
- There is a relationship between PM10 leaf-deposited data and PM10 air concentration values both from ECS records and TAPM model predictions;
- The local steel plant has an impact on the urban area air quality, and it is possible to quantify it through the three adopted approaches, in particular with PM10 deposited data.

The paper is organized as follows: Section 2 presents an overview of the case study and methods used. Section 3 focuses on the achieved outcomes. In Section 4, the results are discussed and analyzed. Finally, in Section 5, lessons learned, and future activities are presented.

## 2. Materials and Methods

Three main data sources for the PM10 assessment were used: dispersion modeling results, ECS monitored data, and leaf deposition values. TAPM was applied to simulate PM10 levels for one year. The ECS, among others, recorded the hourly PM10 concentration values, and their data were used to describe PM dynamics across the urban area and to compare with TAPM simulated values. Leaf deposition data were collected for 55 weeks: for five periods, from January 2012 to January 2013, leaves were sampled from the same trees (one species: *Quercus ilex*), in four different districts of the urban area.

### 2.1. Study Area

The city of Terni is located in the southern Umbria region, in Central Italy (42°34′ N; 12°39′ E). It is a relatively small town, consisting of around 112,000 inhabitants, and is characterized by an industrial history and economy. The "Thyssen Krupp AST", a large steel factory, was founded at the end of the 19th century. Two more recent chemical–industrial areas are located close to the city center.

Because of the intensive industrial activities and the location of the city within a plain surrounded by mountains, atmospheric pollution is a major local issue. Very high PM10 concentrations occur throughout the year, particularly in winter. The PM10 daily limit value established by the European Air Quality Directive (50 μg·m$^{-3}$) was exceeded on more than 70 days in 2012, the reference year for this study. Still, nowadays, Terni is facing exceedances to this limit value (Regional Agency for Environmental Protection Umbria–ARPA Umbria).

The study area includes four districts: Borgo Rivo, Verga, Carrara, and Le Grazie (Figure 1B). These areas were chosen due to the presence of an ECS managed by the ARPA agency.

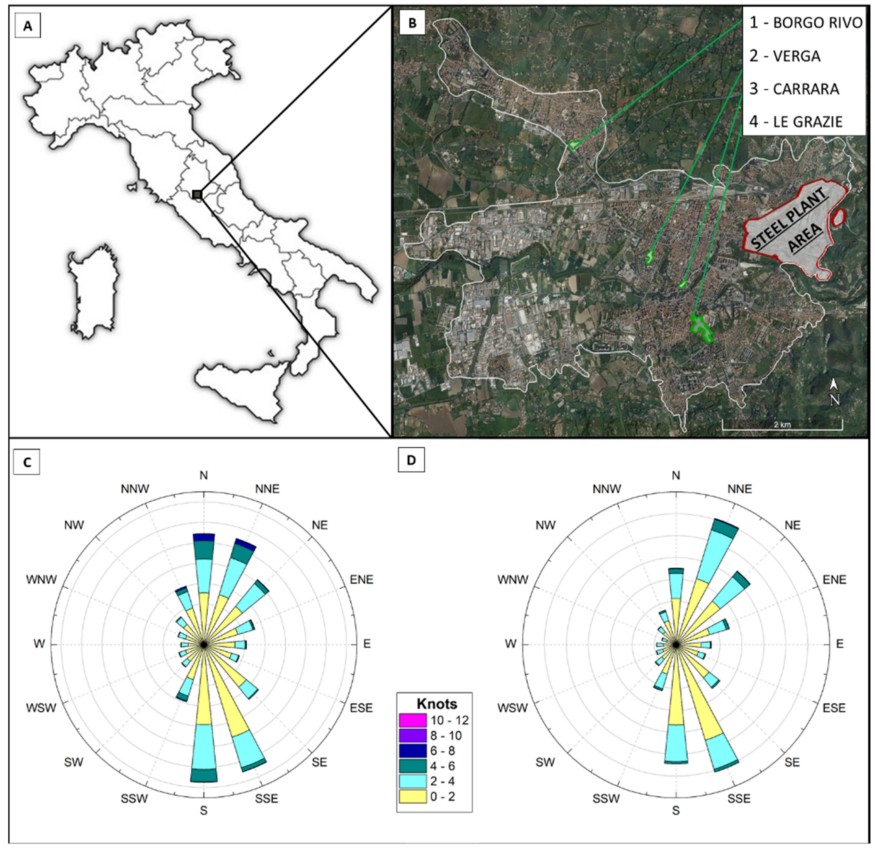

**Figure 1.** Study location in Italy (**A**); city of Terni and sampling areas with reference environmental control stations (ECS) and steel plant area position (**B**); wind-rose of prevailing winds as seasonal averages for autumn-winter, AW (**C**); and spring summer, SS (**D**) periods.

Wind data from the closest weather station, in the city of Narni (Narni Scalo meteorological station: coordinates WGS84: 42.5513888, 12.544166642—source: https://annali.regione.umbria.it/) (access on 18 November 2021), 6 km far from Terni city center and located in the same geomorphological structure of the study area [14], was used to understand the wind patterns in the region. Figure S1 (see in the Supplementary Materials) shows the exact position of the meteorological station that recorded the analyzed data in this study. The dominant winds in the Terni city are from North and North-East and South and South-East. During the Autumn–Winter period, northern and southern direct winds

are stronger and more frequent than during the Spring–Summer period (Figure 1C,D). According to the location of the steel production industrial area, there is a chance for this to have an impact on the air quality of the city center.

The study area was represented and classified through GIS analysis (QGIS 3.16 and previous versions—QGIS.org, 2021. QGIS Geographic Information System. QGIS Association. http://www.qgis.org) (access on 18 November 2021). As the main criterion, the land use activities were considered. Four main classes that embody the urban pattern landscape and five sub-classes were assumed, for a total of 7 classes: (i) Residential (areas including residential buildings, with a growing presence of green areas surrounding, described in % as follows: 1, <10%; 2, between 15 and 25%; 3, >50% of green areas). (ii) Industrial: 1, all industries except the steel factory; 2, steel factory Thyssen Krupp—AST). (iii) Agriculture. (iv) Nature. Figure 2A shows the distribution of the 7 land use classes for the city of Terni. The major roads in the city were also identified and classified depending on traffic counts provided by the municipality of Terni. Three classes were defined based on the number of vehicles per day (#Vd$^{-1}$): (i) heavy traffic roads (>10,000 Vd$^{-1}$); (ii) medium traffic roads (between 5000 and 10,000 Vd$^{-1}$); (iii) light traffic roads (<5000 Vd$^{-1}$). Figure 2B displays the main Terni roads per class. The whole classification of potential fix (land use) and mobile sources (traffic roads) of PM is resumed in Table 1.

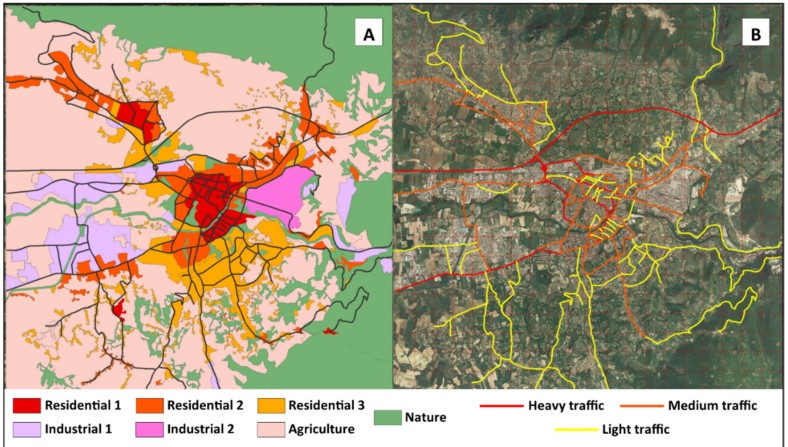

**Figure 2.** Land use characteristics of the study area. (**A**) GIS seven classes: Residential 1, green areas <10%; Residential 2, green areas $\simeq$20%; Residential 3, green areas $\simeq$50%; Industrial 1; Industrial 2, steel plant; Agriculture; Nature. (**B**) Traffic roads per traffic class: 1—heavy; 2—medium; 3—light.

**Table 1.** Study area classification through land use activities. For Agriculture and Nature classes the provided descriptions follow the Corine Land Cover nomenclature.

| | Fix Sources (Land Use Areas) | | | | Mobile Sources |
|---|---|---|---|---|---|
| **Class** | **Residential** | **Industrial** | **Agriculture** | **Nature** | **Traffic Roads** |
| **Sub-Class description** | Residential 1 green areas <10% | Industrial 1 (other industries) | Arable lands; Permanent crops; Heterogeneous agricultural areas | Mixed forests; Broad-leaved forests; Shrubs/herbaceous vegetation; Water courses | (1) Heavy (>10 K Vd$^{-1}$) |
| | Residential 2 green areas 15–25% | Industrial 2 (Steel Plant) | | | (2) Medium (5 K < Vd$^{-1}$ < 10 K) |
| | Residential 3 green areas >50% | / | | | (3) Light (<5 K Vd$^{-1}$) |

## 2.2. Chemical Transport Model Simulations and Validation

TAPM, which was developed under Australia's Commonwealth Scientific and Industrial Research Organization [15,16], was selected to perform the air quality simulations over the study region.

TAPM is a 3D Eulerian incompressible, non-hydrostatic, primitive equations model, which uses a terrain-following coordinate system [16]. It is composed of two modules that predict meteorology and air pollution concentrations. This modeling system has already

been extensively applied worldwide, exhibiting good agreement when compared against observational data [17–21].

TAPM uses the coarse resolution (~80 km) global meteorological data output from The Australian Community Climate and Earth-System Simulator (ACCESS) model, developed by the Australian Bureau of Meteorology (BOM) to downscale in higher resolution to any region of interest in the world. The ACCESS models, as developed by The Australian Bureau of Meteorology (BOM), are based on the UK Meteorological Office's Unified.

For the topography TAPM's standard data sets, we included global terrain height with a resolution of 1 km, as well as vegetation and soil types with a resolution of 5 km and 1 km, respectively. The data are available from the US Geological Survey Earth Resources Observation Systems (EROS). Due to the spatial resolution used ($0.5 \times 0.5$ km$^2$), no street canyons configurations were considered.

TAPM was run on chemistry mode with sulfur and fine particle chemistry. The gas-phase is based on a semi-empirical mechanism entitled the Generic Reaction Set (GRS), including 10 reactions for 13 species. To simulate the air quality for the year 2012, TAPM has been set up on four nested domains with a horizontal resolution of 32, 8, 2, and 0.5 km side-length, centered on the city of Terni, Italy. The inner domain covers an area of $10 \times 10$ km (Figure 3). Background concentrations used by the model were obtained by estimating the annual average of the background air quality values measured by the monitoring sites in the study region during 2012.

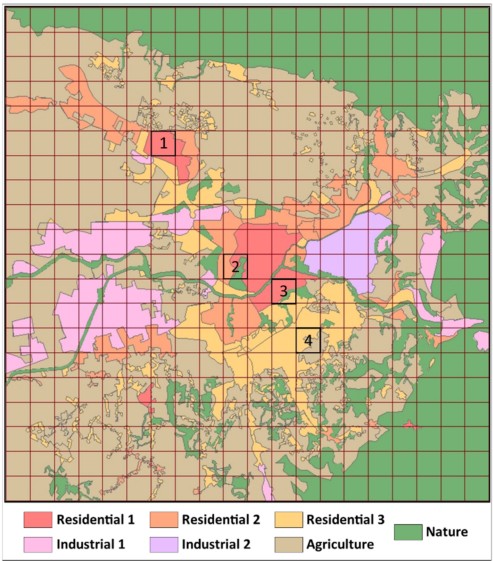

| Residential 1 | Residential 2 | Residential 3 | Nature |
| Industrial 1 | Industrial 2 | Agriculture | |

**Figure 3.** Inner domain and cells grid (1 cell = $0.5 \times 0.5$ km). The map shows the land use composition within each cell through land-use-related colors, and ECS positions within the grid cell (1, Borgo Rivo; 2, Verga; 3, Carrara; 4, Le Grazie).

Air pollutant emissions, namely PM10, sulfur dioxide (SO$_2$), nitrogen oxides (NOx), and volatile organic compounds (VOC), from both fixed and mobile sources were considered for the simulation. Annual data from the ARPA Umbria atmospheric emission inventory (http://apps.arpa.umbria.it/inventarioemissioni/Dati.aspx) (access on 7 January 2022), for 2012, for each pollutant and activity sector, were spatially and temporally disaggregated using a top-down approach to obtain the required resolution for the selected simulation domain. Table 2 shows the reported PM10 emissions per Selected Nomenclature for Air Pollution (SNAP) categories. For the emissions' temporal disaggregation on a daily basis, the weight attributed to each hour of the day by the activity sector followed the standard profile provided by TAPM. The disaggregation of emission data along the 24 h per SNAP (macro-sector) that has been implemented is shown in Figure S2. No distinction is made

for weekdays and weekends. Regarding the spatial disaggregation, all SNAPs included within each cell give their contribution to the relative cell in terms of emissions.

**Table 2.** Total PM10 emissions (tons) per SNAP macro-sector, as provided by the ARPA Umbria for the Terni municipality.

| ID | SNAP Macro-Sector | Emissions PM10 ($t \cdot y^{-1}$) |
|----|-------------------|-----------------------------------|
| 1 | Power plants | 1.32 |
| 2 | House heating | 393.42 |
| 3 | Industrial combustion | 2.64 |
| 4 | Production process (steel) | 99.39 |
| 5 | Solvents | 6.07 |
| 6 | Transports | 74.14 |
| 7 | Other mobile sources | 2.65 |
| 8 | Waste management | 0.12 |
| 9 | Agriculture | 6.27 |
| 10 | Nature | 0.05 |
| | **TOTAL** | **586.08** |

Except for SNAP6 and SNAP7 emissions, which were processed as line source emissions, all the other sectors were treated as area source emissions and their values were converted in gridded surface emissions through disaggregation of the emission inventory.

According to the PM10 emissions data (Table 2), the most important sector was "house heating". This sector's total emissions were distributed among the different sub-residential classes previously stated (Figure 2A): 45% for Residential 1; 35% for Residential 2; and 20% for Residential 3. Moreover, emissions from this "house heating" sector were considered only for the winter weeks (expressed as weeks of the year—WoY): 01–15 and 42–55.

Production process emissions (SNAP4), as well as SNAP1 and SNAP3 industrial emissions, were allocated to the grid cells that were classified as industrial areas. The other minor emission sources (solvents, waste management, and agriculture) were also distributed, taking into consideration the spatial distribution of land-use classes. For the model run, a total emission value for each cell of the grid used ($t \cdot y^{-1}$ per cell) was obtained.

The third higher emission category (SNAP 6—mainly related to road traffic emissions) was processed as a line source. The total emissions of "transports", and also "other mobile sources" sectors, were allocated according to the road classes, which were defined based on the number of vehicles per day (Figure 2B): 40% for class 1, 20% for class 2; 15% for class 3. The residual 25% was distributed on the whole grid, considering the small roads not evidenced in the map (Table 1).

### 2.3. PM10 Concentration and Deposition Measurements

Air quality data from the ECS network in Terni were used for the model validation results. Four ECS were included in the study, one per district, with the following coordinates: Borgo Rivo (42.582, 12.623); Verga (42.561, 12.639); Carrara (42.561, 12.651); Le Grazie (42.550, 12.650). ECS positions within the inner domain grid are shown in Figure 3.

Regarding PM10 deposition on leaves, the present research reports a part of this data and integrates it with new information in an already published article [22]. In the previous research, three sampling periods for the year 2012 were considered and compared. Leaves from holm oak trees (*Quercus ilex*, L. 1753) were collected on the following dates: 11 January, 23 August, and 15 October 2012. In the present study, two more sampling periods were added: April 2012, sampling day 12, and January 2013, sampling day 15. Leaves were gathered from the same trees of the previous campaign. Branches from three trees in each district were sampled (total 12 trees) per period. On each branch, the number of leaves

ranged between 50 and 150. The analysis was performed on different aged leaves; thus, they were exposed to the air for at least the three months. Three replicas per tree were allowed to have a mean value representative of the whole canopy. To obtain PM10 deposition data on leaves, the gravimetric technique of vacuum filtration (V/F) was adopted. Sampled leaves were washed in micro-distilled water and then the solution was filtrated through filters at different porosity. This process has been often employed in the quantification of PM10 deposited on leaves and/or similar structures [23–26]. Further information regarding the V/F process, the specific steps, and the materials used also in this study are described in detail by Sgrigna et al. (2015) [22].

*2.4. PM10 Datasets and Relationship with Steel Plant: Steel Factor Coefficient*

As mentioned above, the steel plant is one of the most important fixed sources in the study area. The steel production process, with 99.39 t·y$^{-1}$ (Table 2), contributes to over 52%, during the spring–summer period, and 11%, during the autumn–winter period, of the total yearly PM10 emissions (Figure S3). Furthermore, previous studies confirmed this influence and investigated the effect of the steel plant area across distinct districts within the city [27]. For approximately 8 months per year, the steel plant represents more than half of the total PM10 emitted in the city.

Thus, regarding the steel plant, we assumed that: (i) it is a reference-polluting source; (ii) its effect is noticed across time (sampling periods) and space (districts); (iii) its relationship with districts can be used to test and compare the three datasets. Therefore, a new factor, here defined as "steel factor" (SF), which is a distance and wind-based value directly related to the steel plant area, was introduced. The factor was created to provide the analysis of a source–receptor relationship. The SF is a coefficient, and it is directly related to steel plant position through wind data and the relative distances of each ECS from the steel plant.

To compare the three different datasets through the SF and considering the effect of the steel plant as a PM10 source on the different districts and seasons, a regression analysis was performed. SF was the predictor (independent variable) for the linear regression, while the three compared PM10 datasets were considered as dependent variables. This analysis allowed us to verify the PM10 datasets' reliability and, in the meanwhile, to compare the concentration data (ECS and TAPM), expressed as µg·m$^{-3}$, directly with leaf surface-related data (leaf PM10 deposition data), expressed as µg·m$^{-2}$. The SF allowed investigation of, in a single analysis, the spatial relationship between PM10 dispersion and deposition. The SF formula is shown in Equation (1), and resulting values are expressed in km$^{-1}$.

$$\text{SF}\left(\text{km}^{-1}\right) = \left(\frac{1}{D}\right) \cdot dW\% \tag{1}$$

The SF was obtained by combining the position of each district relative to the steel plant area (Figure 1) and the winds blowing from the quarter where the steel plant rises, in the direction of each district. The inverse value of distance (km) of each sampling area from the steel plant (1/D) was related to the percentage of winds ($dW\%$—"downwind") coming from the factory zone. For wind data, hourly records from the same dataset described in Section 2.1 were considered. To calculate the $dW\%$ values, all wind hours from the interested quarters, for the previous 3 months before every sampling campaign, were summed, then expressed as a percentage (where 2232 number of hours in 3 months is the 100%), on a 0–1 scale. The interested wind quarters for each district are grouped as follows: S, SE for Borgo Rivo; NE, E, SE for Verga and Carrara; N, NE for Le Grazie. In Figure 1 the relative position of each district with respect to the Steel Plant area is shown. To determine adequate wind directions, the wind rose (degrees scale) was divided into 8 fractions: North (N), 342.5°–359° and 0°–22.5°; North–East (NE), 22.5°–67.5°; East (E), 67.5°–112.5°; South–East (SE), 112.5°–157.5°; South (S), 157.5°–202.5°.

A total of 16 SF values was calculated with a single SF per sampling area (district) in every considered season. All values recorded to calculate the relative SF, namely the

distance from the steel plant and its inverse (1/D) and the percentage of downwind hours in the period before every sampling campaign (*dW*%) are reported in Table 3.

**Table 3.** Steel Factor (SF) values obtained per each district for the four sampling periods. SF is the result of the combination of distance from the factory and the percentage of winds (percentage of hours) blowing from the factory area quarter: S, SE (Borgo Rivo); E, NE, SE (Verga and Carrara); N, NE (Le Grazie).

| Sampling Month | Sampling Area | *dW*-Winds from Factory (%, 0–1) | D-Distance from Factory (km) | 1/D-Distance Inverse | (SF) Steel Factor (km$^{-1}$) |
|---|---|---|---|---|---|
| | *Borgo Rivo* | 0.04 | 3.83 | 0.26 | **0.011** |
| **Apr** | *Verga* | 0.12 | 1.98 | 0.51 | **0.063** |
| | *Carrara* | 0.12 | 1.33 | 0.75 | **0.094** |
| | *Le Grazie* | 0.36 | 1.55 | 0.65 | **0.232** |
| | *Borgo Rivo* | 0.05 | 3.83 | 0.26 | **0.013** |
| **Aug** | *Verga* | 0.12 | 1.98 | 0.51 | **0.059** |
| | *Carrara* | 0.12 | 1.33 | 0.75 | **0.088** |
| | *Le Grazie* | 0.31 | 1.55 | 0.65 | **0.203** |
| | *Borgo Rivo* | 0.06 | 3.83 | 0.26 | **0.016** |
| **Oct** | *Verga* | 0.16 | 1.98 | 0.51 | **0.078** |
| | *Carrara* | 0.16 | 1.33 | 0.75 | **0.117** |
| | *Le Grazie* | 0.33 | 1.55 | 0.65 | **0.213** |
| | *Borgo Rivo* | 0.05 | 3.83 | 0.26 | **0.013** |
| **Jan** | *Verga* | 0.15 | 1.98 | 0.51 | **0.077** |
| | *Carrara* | 0.15 | 1.33 | 0.75 | **0.114** |
| | *Le Grazie* | 0.36 | 1.55 | 0.65 | **0.232** |

For ECS and TAPM values, the averages recorded for the week before every leaf sampling day were considered, while for leaf deposition, data reported in Table S1 were used. SF comparison with PM10 data was performed four times, once per leaf sampling period, as described in Section 2.3: April 2012, August 2012, October 2012, and January 2013. January 2012 was excluded due to its too-short period (less than the 3 months to evaluate the seasonal variability) that occurred before the first sampling.

## 3. Results

### 3.1. PM10 Seasonal Variability and Data Comparison

Based on the hourly PM10 concentration values estimated by TAPM, seasonal averaged patterns were estimated. Figure 4 presents PM10 concentration maps for the two main considered seasonal periods (Autumn–Winter—AW, and Spring–Summer—SS).

For the AW period, higher values of PM10 concentration were obtained over the city center (Figure 4A), the highest peak of these seasonal values almost reached 50 μg·m$^{-3}$. Meanwhile, for the SS period, a different PM10 pattern was estimated, with higher values from the eastern to the western side of the domain, indicating the potential effect of the steel plant emissions (Figure 4B). Differently from the AW values, SS averages showed lower peak values (~40 μg·m$^{-3}$). However, larger minimum concentrations were obtained for SS; the minimum value predicted was 20 μg·m$^{-3}$, while for AW this value was 10 μg·m$^{-3}$.

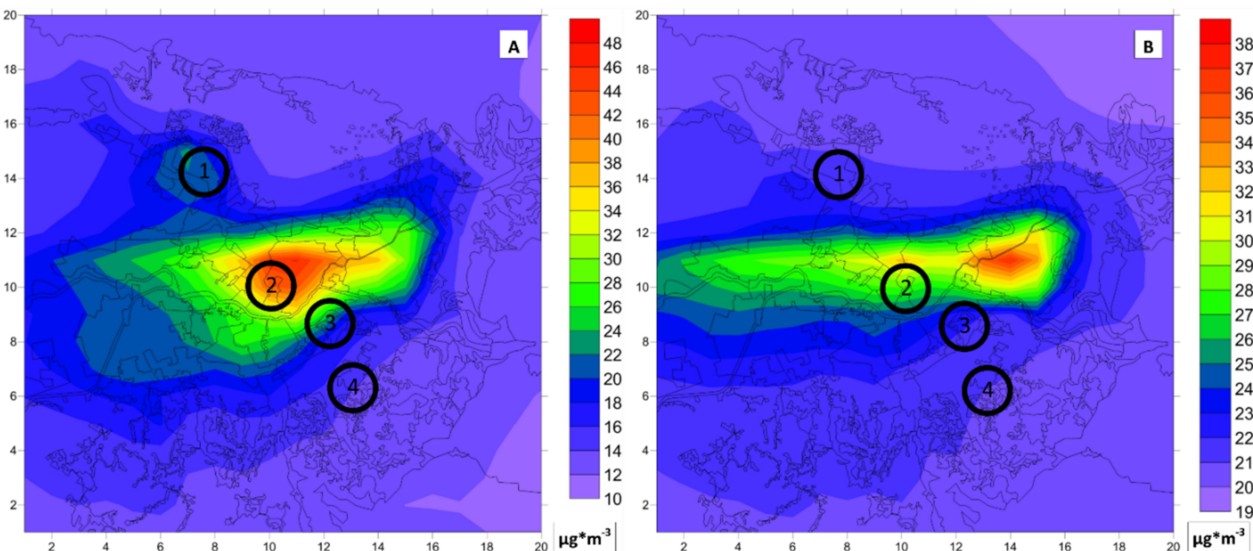

**Figure 4.** TAPM simulated PM10 concentration levels patterns for Autumn-Winter AW (**A**) and Spring-Summer SS (**B**) periods. Numbered circles show sampling areas and ECS locations.

The TAPM simulated concentrations, by grid cell, were compared with ECS measured values. To observe periodic trends and macroscopic differences, hourly data were averaged at different time scales: daily (data not shown), weekly (Figure 5), and seasonally, considering the two periods, AW and SS (Figure S5). Weekly data averages are presented in Figure 5, where three records are contrasted: model predictions, ECS data, and leaf-deposited PM10.

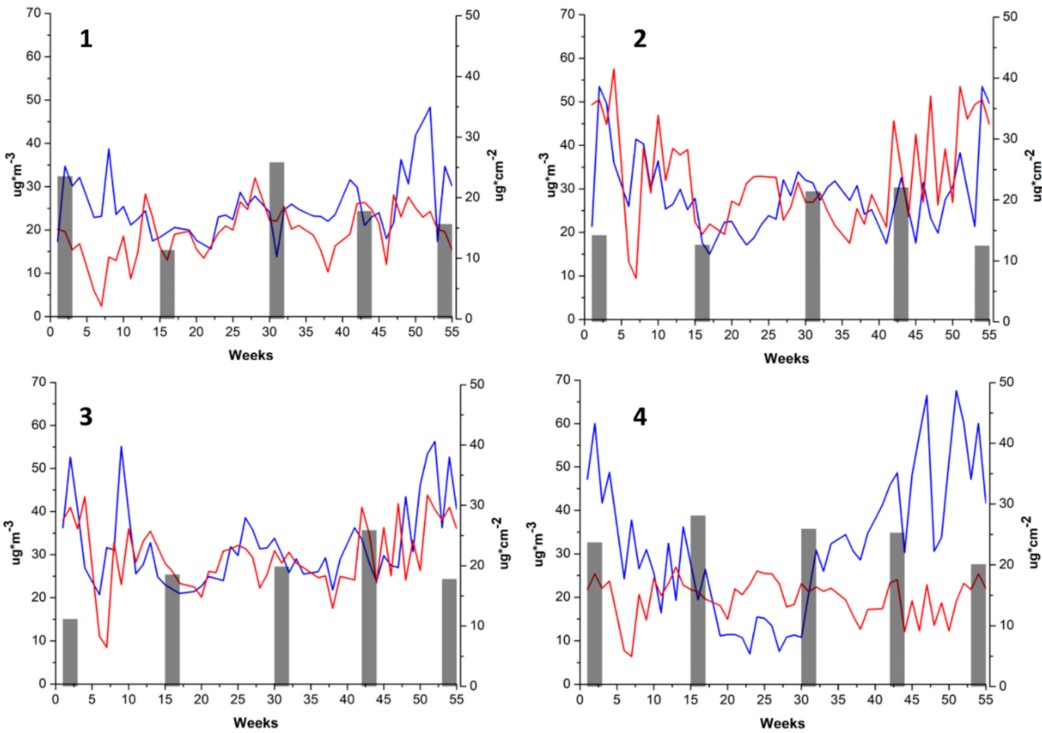

**Figure 5.** Comparison between TAPM (red line) and ECS–ARPA (blue line) PM10 values (weekly averages) in the four districts: (**1**) Borgo Rivo, (**2**) Verga, (**3**) Carrara, (**4**) Le Grazie. Overlap with PM10 deposition data ($\mu$g cm$^{-2}$) on sampled leaves (*Q. ilex*) in five periods (grey istograms): 11 January 2012, 12 April 2012, 23 August 2012, 15 October 2012, and 15 January 2013.

In the comparison between measured data and model outputs, a general similarity was observed, both for average values and for trends, except for Le Grazie ECS (Figure 5,

graph (4)). The global fit between the two datasets is also evidenced by some specific sudden events and a strong drop in PM10 concentration in all four districts: namely for the WoY 6, 36, and 37 (Figure 5, all graphs). These occurrences are directly linked to remarkable rain episodes, as evidenced by the rain dynamic of the year in the study area (Figure S4). This rain effect was also observable in the last period of the year: the whole period after the 40th week was characterized by a continuous sequence of peaks and drops, both in ECS and in TAPM data. This is evidenced in all four districts.

Nevertheless, some differences between the model and measured data characterized two districts: Le Grazie firstly, and Borgo Rivo secondary, both for the AW period. Two separated tendencies were observed: stations located in the central part of the city, namely Carrara and Verga (Figures 3 and 4, points 2 and 3) show similar averages; while for Borgo Rivo and Le Grazie districts that are farther from the city center (Figures 3 and 4, points 1 and 4), the model-simulated values are lower, and there is an underestimation of the concentration values (Figure 5). This condition was noted in both districts for the AW period, but with a larger gap for the Le Grazie district. Furthermore, for Borgo Rivo, differences were observable for the periods included within WoY 01–12 and 48–52, while for Le Grazie, a longer period presented this underestimation (between WoY 01 and 15, and between 31 and 52 WoY). Moreover, in this district, for the remaining weeks, which were strictly the summer period (16–30 WoY) there was an inversion of the situation previously described: the model seems to overestimate the PM10 air concentration values as recorded by the ECS (Figure 5, graph (4)).

On the other hand, for Verga and Carrara, no remarkable differences are evidenced, both for weekly and for seasonal averages. In the seasonal comparison, the boxplots are overlapped (Figure S5). It is to be noted, however, that for the Verga district, the model is slightly overestimating PM10 concentrations for the AW period, mainly for the 42–52 WoY (Figure 5, graph (2)).

Finally, regarding the seasonal comparison, we point out that the AW period averages were generally 20% higher than the SS averages. However, the spatial analysis of these trends indicated that in the more peripheral districts (Borgo River and Le Grazie), ECS records show higher values during AW than SS periods (+17% in Borgo Rivo and +47% in Le Grazie), while TAPM predictions evidence an opposite behavior (−5% in both districts).

Modeled and measured concentration data were also collated with *Q. ilex* leaves PM10 deposition data acquired by laboratory analysis. The two records are not directly comparable, since they expressed two different types of data (TAPM and ECS, unit per air volume, $\mu g \cdot m^{-3}$; Leaf deposition, unit per leaf surface, $\mu g \cdot cm^{-2}$). Nevertheless, a first general tentative of comparing data confirms similarities between PM10 air concentration and deposition. The higher levels of PM10 recorded by ECS in the Le Grazie district for the AW period were confirmed. The average value of PM10 deposition on leaves for all sampling periods in the Le Grazie district (24.4 $\mu g \cdot cm^{-2}$) was about 25% higher than the average of the other districts. Likewise, for the same period, ECS records in Le Grazie were 23% higher than the average of the other districts (Table S1).

Regarding PM10 on leaves, it is noteworthy that for all districts the sampling of Jan 13 shows lower values than for the previous ones, with an average loss of 20.1% of deposited PM. Furthermore, excluding the Le Grazie district, it was possible to identify a general common pattern in the accumulation: a trend of a growing quantity of PM seemed to have accumulated from WoY 2 to 30, or up to the 43 WoY in the case of Verga and Carrara districts. All detailed values, averages, and relative standard errors are reported in Table S1.

*3.2. Cross Analysis and Steel Plant Impact Evaluation*

Through the newly introduced SF, the three PM10 datasets were directly compared together. Figure 6 shows the obtained relationship: the three PM10 datasets were considered as dependent variables, while seasonal SF values were the predictors. Each graph in Figure 6 specifically shows the relationship for one sampling period (April; August; October—2012, January—2013). For every period considered in this analysis, different SF values were

calculated. Its variability among districts was mainly related to their relative distance from the steel plant, while the variability across the four periods was strictly related to downwind hours percentages (*dW*%).

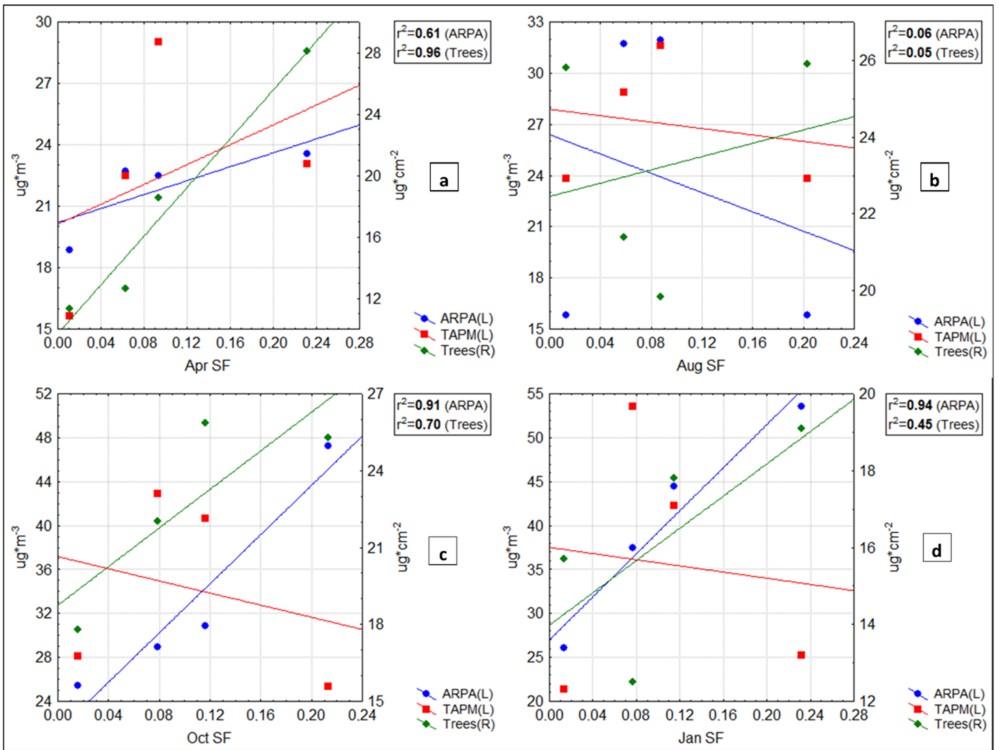

**Figure 6.** Scatterplot relationships between steel factor (SF) and: ECS (ARPA data, blue signs), TAPM outputs (red signs), and leaf deposition data (green signs). Different SF values were associated with each district for the 4 sampling periods (detailed values are shown in Table 3). The periods represented are: (**a**) April 2012; (**b**) August 2012; (**c**) October 2012; (**d**) January 2013. Leaf PM deposition as compared with air concentration data (ECS and TAPM model prediction), based on the steel factor (SF) independent variable.

Thus, consistent with the wind data variability and the different relative distances of each ECS from the steel plant area, SF significantly varied in the four considered sampling periods. Nevertheless, as reported in Table 3, their values were always in a similar range in the different periods for each sampling area. The lowest values were found for Borgo Rivo (0.011 < SF < 0.016), while the highest values were recorded for Le Grazie district (0.203 < SF < 0.232), while for Verga and Carrara, SF adopted mostly "intermediate" values, which included between 0.059 (Verga, August) and 0.117 (Carrara, October). The relative SF stability over time and space showed the same districts' order across the four periods: Borgo Rivo > Verga > Carrara > Le Grazie, with just slight variations within each district, mostly depending on *dW*% variability.

When SF was correlated to the three PM10 datasets, the spatial relationship related to the steel plant was confirmed for three periods, for ECS and Leaf deposited PM10 data, reported respectively as "ARPA" and "Trees", and not for TAPM (Figure 6). Therefore, for correlations with TAPM, the coefficient of determination values ($r^2$) is not shown in the graphs: values are always set below significant correlation values ($r^2 < 0.2$).

ECS records (ARPA) were strongly related to SF values for three out of the four periods: namely April ($r^2 = 0.61$), October ($r^2 = 0.91$), and 13 January ($r^2 = 0.94$) (Figure 6a,c,d). On the other hand, TAPM predictions did not fit with the regression analysis and never showed a significant relationship with the coefficient SF. Nevertheless, regarding TAPM, it is noteworthy that the main evidenced discrepancies mostly depended on Le Grazie district values, as similarly shown in Section 3 with the ECS ground data comparison.

Similar to the ECS records, Leaf deposition data were also connected with the SF coefficient, even if with weaker relations for the same periods (Figure 6a,c,d): April ($r^2$ = 0.96); October ($r^2$ = 0.70); 13 January ($r^2$ = 0.45). By the regression analysis, leaf deposited PM10 on leaves showed a stronger correlation with SF for the "intermediate" seasons of the year: April and October samplings. On the contrary, for the "extreme" seasons, the relationship was weaker, for full winter (January 2013 period), or not verified, for full summer (August period). It is also outstanding the total absence of relationships, in all datasets for the August sampling.

This section may be divided by subheadings. It should provide a concise and precise description of the experimental results, their interpretation, as well as the experimental conclusions that can be drawn.

## 4. Discussion

Differences in the PM10-modeled concentrations when autumn–winter (AW) and spring–summer (SS) periods were compared were already expected. The higher observed peak values for the AW period were driven by the weight of the house heating source, which represents almost 80% of the winter emissions of PM10 (Figure S3). House heating in the urban environment is recognized to be one of the main PM10 sources during wintertime (e.g., [28,29]). We also point out the lower minimum values predicted by TAPM in the same period, mainly estimated for the most peripheral and rural parts of the domain. The model considered the effect of rain as natural air cleaner, which, in temperate climate areas, is usually much more abundant during the AW period.

Similarly, ECS records evidenced the higher concentration of PM10 during the whole AW period (1–15 and 42–55 WoY), in the four districts. This condition was observable mainly in Le Grazie, and also in Borgo Rivo and Carrara, while it was less evident in the Verga district, which shows peaks only for the periods between 1–5 and 50–55 WoY (Figure 5 and Figure S5). The Verga district area, located in the central part of the city, should be more influenced by the house heating effect; nevertheless, as described by Salma and Maenhaut (2006) [30] for the city of Budapest, the central part of an urban area is influenced by several pollution sources. Furthermore, it is noteworthy how the seasonality was more evident in ECS than TAPM. This condition is probably connected to the temperature inversion, a phenomenon that could easily occur in a significant topography such as Terni valley, and which is detected by ECS and not evidenced by TAPM.

Regarding PM10 deposition on leaves, this seasonal pattern connected with the AW period was not evidenced; on the contrary, there was a growing quantity of PM10 deposition during the SS periods and a significant drop of PM10 quantity in the last sampling period (January 2013). This opposite pattern is explainable by rain dynamics, which wash off the deposited PM during the AW period, while a longer dry season (as during the SS period) facilitates leaf PM10 deposition. Indeed, the rain-cleaning effect on leaves is a phenomenon that has been observed and described in detail by previous researches [31,32] and especially by Popek et al. (2019) [23] who underline the rain effect mostly on larger particles.

The differences in the spatial patterns derived from TAPM predictions and ECS data could be related to input data allocation. In the AW period, an overestimation of PM10 concentration for TAPM was observed in the Verga district, while there was an underestimation in Le Grazie, Borgo Rivo, and partially also in Carrara districts. This situation occurred mainly after the 45th week (Figure 5). A different emission allocation during the disaggregation process in the three Residential classes, highly related to house heating sources, would affect the estimated PM10 values and probably mitigate this disagreement.

Nonetheless, the Le Grazie district has to be highlighted, due to the relevant differences between model results and measurements. As described in Section 3, for this sampling area, the PM10 deposition on leaves was much higher than in the other districts, and ECS values showed the largest discrepancy with TAPM predictions. Furthermore, the higher ECS levels, when compared with TAPM predictions, were widely extended beyond the AW period. The above-mentioned interval was characterized by twelve weeks (included

between the WoY 30 and 42), covering the full summer and the early autumn, and thus house heating was not the main influencing source for this period. Therefore, for a large period, the house heating influence and the uncertainty in the disaggregation process have to be excluded.

Le Grazie district has been mentioned as being directly influenced by the steel plant. In Sgrigna et al. (2016) [33], the chemical and dimensional PM10 composition is accurately described, by comparing Borgo Rivo and Le Grazie. Particulate matter for this district is reported as being finer and with an iron percentage composition four times higher than that of Borgo Rivo. A simplified version of data described in that research was resumed and reported in Figure S6. Also, Massimi et al. (2020) [34] confirm, in a detailed study of the PM10 spatial mapping for the city of Terni, the steel plant effects on the Le Grazie district.

We also highlight that the above-mentioned peaks of particle concentration represent weekly averages of PM10. Thus, the districts showing higher peak presence are also affected by a potential chronic and sub-chronic citizens' exposure to PM10. This health issue becomes significantly relevant for Le Grazie district, where for the 29% of the recorded WoY, the PM10 average is >40 $\mu$g·m$^{-3}$.

The steel plant is an influencing source in the Le Grazie area that was not identified by the TAPM simulation, probably because of emissions and wind data. Different wind directions as input data could change the PM10 higher values from the central areas (e.g., Verga and Carrara districts) to the peripheral southern district of Le Grazie. However, this change would also involve PM10 concentration drops in the central districts, where ECS and TAPM were generally well related. An improvement in the steel plant PM10 TAPM inputted emission values should be considered. In fact, for the present study, data were based on the available emission inventory for industrial emissions (Table 2, SNAP 4), since no more detailed data was available. As stated by Büns et al. (2012), and Zheng et al. (2009) [35,36] air pollution inventories used for model predictions could present important uncertainties that can directly influence the model outputs.

Moreover, PM10 resuspension has also to be considered. Mangia et al. (2020) [37] report the significant influence of resuspension from steel plant mineral parks on PM10 concentration within the districts downwind to the factory area in the city of Taranto. The city of Taranto, with its steel plant, in the Apulia region (southern Italy), is an urban–industrial area with comparable conditions to the city of Terni. The industrial area and relative annexes are set right alongside the urban area. The districts located downwind to the factory area are observed as being affected not only by PM10 direct combustion processes emissions, but also by re-suspended particles from large mineral deposits of raw material. Similarly, Amodio et al. (2013) [38] also described the connection between considerable PM10 concentration and wind events.

Therefore, the re-suspension effect from the mineral park, which was not considered by the TAPM simulations, could be the main reason for the lower simulated values in the Le Grazie district. The Terni steel plant, indeed, similarly to the Taranto one, presents its mineral park areas. These areas are evidenced in Figure 1 by a relatively small elliptical form (rounded in red) alongside the eastern border steel factory area, and on Figure 3 by the purple patches on the eastern side of the domain grid. Additionally, the traffic emission inventory used in this study did not consider the resuspension phenomenon, which affects PM10 deposition and PM10 concentration values measured by the ECS.

This lack of PM10 resuspension would also explain the different PM10 dynamics observed when the SF was included in the analysis. As shown in Section 3.2, all the TAPM values in the Le Grazie district are in contrast with the wind/distance coefficient (SF) considered in the regression analysis. Lacks or uncertainties of the emission inventory directly influenced the estimated PM10 concentration in the Le Grazie district, but they did not affect the field measurements: for ECS data regression, except for the summer period (August sampling); there is a general agreement with the distance factor from the steel plant. Also, for the second typology of ground data considered in the study, leaf-deposited PM10, there was a similar, though weaker, agreement with SF. Nevertheless, as previously

described, the deposition of PM10 on leaves does not follow the same dynamics of PM10 concentration in the air and is directly influenced by dry and rain periods before sampling. This condition is well evidenced for the January 2013 sampling.

Furthermore, to simultaneously compare the three datasets, the SF was introduced, and the peculiar discrepancies recorded between ECS and TAPM were confirmed by this further source–receptor relationship analysis. The choice of the use of a newly introduced factor, instead of a Gaussian plume, Eulerian grid, or other receptor modeling was mainly driven by the lack of information provided by the local environmental control agency (ARPA Umbria). No specific data of emissions at stack levels for the steel plant was provided.

## 5. Conclusions

The assumptions made for the research and the hypothesis stated in the introduction have been verified. There is a seasonal variability, and it has been evidenced by the three techniques; model predictions are comparable with PM10 in three of the four analyzed districts and, through SF, has evidenced a connection between leaf deposition and ECS data. Finally, the steel factor influence on specific areas of the study area has been highlighted.

Terni's urban environment is characterized by two main PM10 seasonal patterns: higher concentrations in autumn–winter, while lower concentrations were observed for spring–summer. This was confirmed by both ECS (AW averages were 20% higher than SS) and TAPM data. Nonetheless, the pattern followed by the PM10 concentration data does not fully agree with the leaves' surface PM10 deposition data. Long, warm, and dry periods facilitate PM10 deposition and accumulation on leaf blades; periods characterized by stronger rain events negatively affect PM10 leaf deposition, the average recorded loss was 20.1%. The analysis performed by the present research confirmed the general reliability of the TAPM approach to describe PM10 dispersion in the urban environment: a good agreement between measured and modeled datasets was verified. Consequently, the model is a suitable tool for an urban scale analysis, but modeling results can be improved with a more detailed and complete PM10 inventory, in particular by the inclusion of resuspension data.

Moreover, the study evidenced the high risk for human health in specific areas, namely Le Grazie district, deriving from the local steel plant influence. Further, the newly introduced SF allowed us to compare the three datasets and to confirm the effect of the steel plant on specific urban districts ($r^2$ = 0.61–0.94 for ECS; $r^2$ = 0.45–0.70 for leaf deposition). The connection between ground data and the steel plant suggests that part of the model underestimated values could be due to re-suspension from mineral deposits of the same steel plant area. Furthermore, the research provides information on whether and when data from leaf deposition can be used as a proxy for air pollution concentration across the city.

A future perspective of this research is the further comparison of our results with a dispersion model able to evaluate also the dry and wet deposition at ground level, such as e.g., AERMOD (American Meteorological Society/Environmental Protection Agency Regulatory Model Improvement Committee Dispersion Model) can provide. However, the results from this study can already inform researchers, urban managers, and decision-makers on the utility of urban tree leave as bioindicators, as well as models, to assess urban air quality. Finally, the study also confirmed the multiple benefits provided by trees in urban environments: harmful particles are retained for a long time on leaf blades and leaves can be a useful and low-cost tool for environmental analysis control.

**Supplementary Materials:** The following supporting information can be downloaded at: https://www.mdpi.com/article/10.3390/su14020793/s1, Figure S1: Meteorological station localization; Figure S2: Daily profiles per emissions macro-sector (SNAPs); Figure S3: PM10 emissions contribution; Figure S4: Weekly rainfall data; Figure S5: Averages PM10 seasonal data; Figure S6: PM10 elemental composition. Table S1: Detailed values of all datasets results.

**Author Contributions:** Conceptualization, G.S. and C.C.; methodology, G.S. and H.R.; software, H.R.; validation, H.R., A.I.M. and C.C.; formal analysis, G.S. and H.R.; investigation, G.S.; resources, A.I.M. and C.C.; data curation, G.S. and H.R.; writing—original draft preparation, G.S.; writing—review and editing, H.R. and A.I.M.; visualization, G.S., C.C. and A.I.M.; supervision, C.C. and A.I.M.; project administration, C.C.; funding acquisition, C.C. and A.I.M. All authors have read and agreed to the published version of the manuscript.

**Funding:** This research was funded by EUFORICC–PRIN Project (Establishing Urban Forest-based solutions In Changing Cities) grant number Prin 20173RRN2S: Projects of National Interest, founded by the Italian Ministry of Education, University and Research (MIUR). Thanks are due to FCT/MCTES for the financial support to CESAM (UIDP/50017/2020 + UIDB/50017/2020).

**Institutional Review Board Statement:** Not applicable.

**Informed Consent Statement:** Not applicable.

**Data Availability Statement:** Part of data used for the research were found at: http://apps.arpa. umbria.it/inventarioemissioni/Dati.aspx (accessed on 18 November 2021).

**Acknowledgments:** The authors acknowledge the ARPA Umbria for their effort in establishing and maintaining the air quality monitoring sites whose data has been used in this study. A special thanks goes to Caterina Austeri (ARPA Umbria) for her kind collaboration and support.

**Conflicts of Interest:** The authors declare no conflict of interest. The funders had no role in the design of the study; in the collection, analyses, or interpretation of data; in the writing of the manuscript, or in the decision to publish the results.

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
