# Peer review of "Particulate Matter in an Urban–Industrial Environment: Comparing Data of Dispersion Modeling with Tree Leaves Deposition"

_sustainability, doi:10.3390/su14020793_

Round 1

Reviewer 1 Report

I reviewed the article entitled: Particulate matter in an urban-industrial environment: comparing data of dispersion modeling with tree leaves deposition. This paper described the dynamics of PM10 through air pollutant model, environmental control station and leaf deposition data. At the same time, the PM data obtained by different methods were compared from multiple dimensions, and the PM10 emission of steel plant was evaluated by means of data. The whole article has good logic and clear discussions.  Some minor modifications are suggested:

L12 (particulate matter with a ≤10 μm diameter) PM10

L15 (The air pollution model) TAPM

L32 (particulate matter with a ≤10 μm diameter) PM10

L26 The description of modelling and monitoring in keywords is too simple

L30-L93 There are too many abbreviations in the introduction. If certain phrases occur only once or twice, don't use abbreviations. The main abbreviations of this article are TAPM, ECS and SF.

L96, L152 Using the abbreviation TAPM, delete The Air Pollution Model.

L238, L256, L257, L303, L307, L308 t·y-1,Replace all the * between units with · (μg·m-3), For the full text of the unit to carry out a detailed inspection, many units are missing ·

Author Response

Rev # 1

The MS is good at preparation. I think it is suitable for publication in this journal.

The authors thank Rev.1 for the job and consideration.

L12 (particulate matter with a ≤10 μm diameter) PM10

Modified in the text, according to your comment.

L15 (The air pollution model) TAPM

Modified in the text, according to your comment.

L32 (particulate matter with a ≤10 μm diameter) PM10

Modified in the text, according to your comment.

L26 The description of modelling and monitoring in keywords is too simple

According to your comment, the two keywords were modified as follow: “dispersion modelling” and “air pollution monitoring”

L30-L93 There are too many abbreviations in the introduction. If certain phrases occur only once or twice, don't use abbreviations. The main abbreviations of this article are TAPM, ECS and SF.

We significantly reduced the abbreviations, besides the main abbreviations (TAPM, ECS, and SF) we just left two important concepts that we consider useful in the text: Nature-based Solutions (NBS) and Ecosystem Services (ES).

L96, L152 Using the abbreviation TAPM, delete The Air Pollution Model.

               Modified in the text, according to your comment.

L238, L256, L257, L303, L307, L308 t·y-1,Replace all the * between units with · (μg·m-3), For the full text of the unit to carry out a detailed inspection, many units are missing ·

We modified all the reported units at the corresponding lines, and in addition, we verified the units’ accuracy and consistency throughout the whole text.

Reviewer 2 Report

General Comments

This is an interesting and novel analysis.  In general, the analysis does not fully explain or interpret the results that were found.  There are a number of factors that should be brought into the discussion—I mention some below.  In addition, the description is sometimes difficult to follow, and the language can be rambling and imprecise.  Many of the points can be made more concisely.  The authors nicely lay out their hypotheses in the introduction.  I think they can make the case that they have conclusions about each of them.  It would tie things together if the conclusions section specifically addressed each of the hypotheses and what the study showed about them.  I think that information is available from the study, but it is not stated clearly and concisely.  Overall, insufficient information is presented to assess the robustness of the analyses..

Specific Comments

  1. Abstract lines 13-16. These two sentences are repetitive.
  2. Abstract lines 16-19. I find it difficult to decipher what the authors intend to say here.  Some clarification in the verbiage would be helpful.
  3. Line 137 and Figures 2 and 3. The term “unless” might be better stated as “except.” The color scheme in Figure 2 makes it difficult to distinguish between Industrial 1 and Agriculture.  The authors state that the same color scheme is used in Figure 3, but in that figure Industrial 1 and Industrial 2 appear to be the same color, and Agriculture is readily distinguishable.  The authors might consider adjusting the color schemes to make things clearer.
  4. The authors talk about human health concerns in the Introduction and mention the European Air Quality Directive for PM10 of a 24-hr average of 50 ug/m3. This is good information, but I think the authors need to do a better job of putting time scales into perspective.  Human health effects play out over time scales of hours (acute) to months (subchronic) to years (chronic).  TAPM, as used here, appears to be geared towards subchronic to chronic time scales.  Similarly, leaf deposition occurs while the leaves are present on the trees which can be months (subchronic) for deciduous trees to years (chronic) for evergreens.  The authors discuss the timeframe for leaf deposition (e.g., line 446), but that is not put in context with the modeling and measurements.  The discussion and conclusions do not revisit the human health topic that they start with in the introduction.
  5. Materials and Methods. The description of how the model was applied are insufficient to understand the results.  For example, a description of how topography was treated in the modeling analysis is not included (see next comment).  The TAPM documentation says that the model predicts important local flows against a background of larger-scale meteorology provided by synoptic analyses. 
  6. What is the source of the synoptic meteorology in this study? Were data other than that from Narni used?  What was the time scale over which the model was applied—hourly, weekly, annual?  If parsed annual emission were used, can the results be compared to shorter time frame measurements?
  7. Figure 4. The concentration isopleths do not seem to be a good match with the wind roses.  Why is that?  The concentration isopleths seem reasonable given the locations of sources, but it would be useful to see a topographic map (such as in attached file showing Terni and the met site) in the supplemental material in order to better understand the air flows.  Narni, the source of meteorological data, has a rather different alignment to the topography than does Terni.  The authors state that the Narni data were used to understand wind patterns in the area.  I’m not sure that the Narni data will be representative of what is going on in Terni.  The Narni wind rose seems to reflect the ridge of higher elevation running NW to SE there.  That same topography is not seen at Terni.  It would be good to know the precise location of the met station.  The coordinates are only given to the nearest minute, and according to those data it is located on the edge of the Nera River valley that has a N-S orientation as it cuts through the terrain, so channelling of winds is possible.  If the wind rose at Narni is representative one would expect the concentration isopleths to be aligned with higher concentrations to sthe N and to the SSW, but in fact in Figure 4 they are aligned along an E-W axis.

  1. Figure 5.  At all four districts, the pattern seems to indicate lower concentrations in summer and higher concentrations in winter.  Figure S1 shows that home heating is a larger percentage of total PM10 emissions in winter and steel process emissions are a larger percentage in summer.  However, this begs the question of the magnitude of the emissions over time.  Figure 1 gives total PM10 emissions, but there is insufficient information to show how the emissions from each source category changes over time.  This seasonality also raises the question of wintertime temperature inversions in an area with significant topography.  The authors should discuss whether such phenomena are known to occur in the area and how that might affect their findings.  They should also address how effectively TAPM may or may not be in capturing potential inversions. At lines 361-362 there is evidence that TAPM may not be capturing this.
  2. An alternative presentation to Figure S1 is a bar chart showing total emissions by month (if possible) with each bar subdivided by color into the various emissions categories. That would provide a broader perspective on the emissions.  At line 176 the authors state, “Annual data from the ARPA Umbria atmospheric emission inventory … for each pollutant and activity sector, were spatially and temporally disaggregated using a top-down approach to obtain the required resolution for the selected simulation domain.”  More of these data should be shown in the supplemental information. 
  3. Since seasonal differences are seen in the patterns of concentrations, it would be useful to see the wind roses by season. Another helpful piece of information might be a diagnostic wind field model that would incorporate other met stations, including winds aloft, and would give insights into air flows in this complex valley.  That type of information would be very helpful in understanding the results.
  4. The Steel Factor (SF) analysis is an attempt to look at source-receptor relationships. Other options include traditional Gaussian plume modeling, Eulerian grid modeling, or receptor modeling (such as CMB, although it is unclear from Figure S4 whether sufficient speciation data is available).  The authors should justify the advantages the SF approach has in comparison to other approaches.
  5. Figure 6. The description of this figure seems a little muddy and rambling, and I’m not sure I fully understand what the authors are trying to say with this figure.  On first impression it seems to me that there is some agreement between the leaf measurements and measured air concentrations, but the relationships with TAPM are less clear.  Am I understanding this correctly?
  6. I was glad to see that the authors discuss the impacts of rain events and potential wash off. I was thinking of this as I read through the MS and was pleased that was considered.  The authors cite their earlier work on leaf sampling, but it is worth mentioning in the methods section that oaks were sampled, and the time period when the leaves were present on the trees should be specified.
  7. Using total annual emissions from the steel plant (and other sources) and parsing them uniformly over the year, may be an important source of error. It is unclear how the temporally and spatially disaggregated emission (see comment 8) were used in the modeling analysis.  I don’t know the regulations and reporting requirements in this jurisdiction, but in many places stack by stack emissions over time, including stack release parameters, are considered public information that can be obtained with an inquiry.  If that is possible here, it should be done.  It would also be useful if the release parameters were stated.  Industrial sources typically emit through taller stacks, and the emissions can be dispersed before reaching the ground and vegetation.  On the other hand, traffic and household emissions occur near the ground level (and near the breathing zone and near leaf surfaces).  Since it appears that emissions were apportioned to grid cells (and release parameters not specified), how does the absence of release parameters factor into the analysis and interpretation that has been presented?

Author Response

Rev # 2

This is an interesting and novel analysis.  In general, the analysis does not fully explain or interpret the results that were found.  There are a number of factors that should be brought into the discussion—I mention some below.  In addition, the description is sometimes difficult to follow, and the language can be rambling and imprecise.  Many of the points can be made more concisely.  The authors nicely lay out their hypotheses in the introduction.  I think they can make the case that they have conclusions about each of them.  It would tie things together if the conclusions section specifically addressed each of the hypotheses and what the study showed about them.  I think that information is available from the study, but it is not stated clearly and concisely. Overall, insufficient information is presented to assess the robustness of the analyses.

Dear Reviewer #2, thanks for your comment. We appreciate the constructive criticisms you offered, and the weaknesses you raised have been faced and the manuscript was significantly improved following most of your suggestions. If a lack of information could appear in the analysis, it is mostly because during the writing process we choose to limit the extent of the paper and discussion to do not overload the text. Nevertheless, we tried to ameliorate it according to your comments. The language was also enhanced and the descriptions were simplified. Regarding the discussion improvement, we answered in the specific comments section. Finally, we followed the clever suggestion of a better connection between the hypothesis stated in the introduction and their clearer description in the conclusions. We are confident that in this way the assessment of the analysis robustness by the reader has been also enhanced. 

We report here the new paragraph that has been included in the conclusions section: “The assumptions made for the research, and the hypothesis stated in the introduction have been verified. There is a seasonal variability and it has been evidenced by the three techniques; model predictions are comparable with PM10 in 3 on the 4 analyzed districts and through SF has been evidenced a connection between leaf deposition and ECS data. Finally, the steel factor influence on specific areas of the study area has been highlighted.”

Specific Comments

  1. Abstract lines 13-16. These two sentences are repetitive.

The second sentence has been deleted and part of the information merged in the first one.

  1. Abstract lines 16-19. I find it difficult to decipher what the authors intend to say here.  Some clarification in the verbiage would be helpful.

The two sentences have been reworded and more information has been added to clarify the message.

  1. Line 137 and Figures 2 and 3. The term “unless” might be better stated as “except.” The color scheme in Figure 2 makes it difficult to distinguish between Industrial 1 and Agriculture.  The authors state that the same color scheme is used in Figure 3, but in that figure Industrial 1 and Industrial 2 appear to be the same color, and Agriculture is readily distinguishable.  The authors might consider adjusting the color schemes to make things clearer.

The term “unless” has been changed with “except”. Regarding the color scheme in Figure 2 we do not agree about the difficult distinction mentioned by Rev#2. Nevertheless, regarding Figure 3 we added a new color scheme specifically for this figure.

  1. The authors talk about human health concerns in the Introduction and mention the European Air Quality Directive for PM10 of a 24-hr average of 50 ug/m3. This is good information, but I think the authors need to do a better job of putting time scales into perspective.  Human health effects play out over time scales of hours (acute) to months (subchronic) to years (chronic).  TAPM, as used here, appears to be geared towards subchronic to chronic time scales.  Similarly, leaf deposition occurs while the leaves are present on the trees which can be months (subchronic) for deciduous trees to years (chronic) for evergreens.  The authors discuss the timeframe for leaf deposition (e.g., line 446), but that is not put in context with the modeling and measurements.  The discussion and conclusions do not revisit the human health topic that they start with in the introduction.

The present research does not claim to be an epidemiological study, able to deeply face the complex relationship between PM pollution and human health at different time-scale. Nevertheless, the question raised by Rev.#2 is important and we are aware of the high hazard level deriving from PM exposure. Thus, following this suggestion we added this paragraph in the discussion section to include the human health issue: “We also highlight that the abovementioned peaks represent week averages of PM10 concentrations. Thus, the districts showing higher peaks presence, are also affected by a potential chronic and sub-chronic citizens exposure to PM10. The health issue becomes relevant for Le Grazie and Carrara districts, where for the 29% and 18% of the recorded WoY respectively, the PM10 average is >40µg·m-3.” Then, in the conclusions section the following sentence has been added: “Moreover, the study evidenced the high risk for human health in specific areas, namely Le Grazie district, deriving from the local steel plant influence”.

  1. Materials and Methods. The description of how the model was applied are insufficient to understand the results.  For example, a description of how topography was treated in the modeling analysis is not included (see next comment).  The TAPM documentation says that the model predicts important local flows against a background of larger-scale meteorology provided by synoptic analyses. 

Thank you for the comment. The meteorological component of the model is nested within synoptic-scale analyses/forecasts that drive it at the boundaries of the outer grid (Hurley, 2008). For the Terni case study, a downscaling approach was adopted and TAPM was applied with 4 successively nested domains with 20 x 20 regular grid cells, and grid spacing resolutions of 32 x 32 km2, 8 x 8 km2, 2 x 2 km2  and 0.5 x 0.5 km2 side-length centred on the city of Terni, Italy. The setup of the model allowed, therefore, to consider boundary conditions from each larger domain until the smaller one.

TAPM was configured for these nested simulation domains by extracting information from surface information databases, which are provided by the CSIRO Atmospheric Research (Luhar et al., 2004), and include gridded terrain height, vegetation and soil type, sea-surface temperature, and synoptic-scale meteorology.

TAPM uses the coarse resolution (~ 80 km) global meteorological data output from The Australian Community Climate and Earth-System Simulator (ACCESS) model developed by the Australian Bureau of Meteorology (BOM) to downscale in higher resolution to any region of interests in the world. The ACCESS models, as developed by The Australian Bureau of Meteorology (BOM), are based on the UK Meteorological Office’s Unified.

For the topography TAPM’s standard data sets include global terrain height with a resolution of  1 km, vegetation and soil type with a resolution of 5 km and 1 km, respectively. The data is available from the US Geological Survey - Earth Resources Observation Systems (EROS). Due to the spatial resolution used (0.5 x 0.5 km2), no street canyons configurations were considered.

For a clearer explanation to the readers, the last two paragraphs have also been included within the text.   

  1. What is the source of the synoptic meteorology in this study? Were data other than that from Narni used?  What was the time scale over which the model was applied—hourly, weekly, annual?  If parsed annual emission were used, can the results be compared to shorter time frame measurements?

Thank you for pointing out this aspect. The synoptic-scale meteorology is from the Australian Community Climate and Earth-System Simulator (ACCESS) model developed by the Australian Bureau of Meteorology (BOM) and is used to downscale in higher resolution, such as 0.5 km by 0.5 km, to any region of interest in the world. Synoptic scale meteorology datasets are available every 6 hours (00,06,12,18).

In addition to meteorological data calculated by the meteorological component of TAPM, this air pollution modelling component requires emission values, which have to be inputted by the user and are usually based on available emission inventories. The emissions are provided on an hourly basis by disaggregating temporally the annual emissions using profiles according to the emission activity sector.

Background air pollutants concentrations were also provided to TAPM based on air quality available data for Terni.

Technical details of the model equations, parameterizations, and numerical methods are described in the Technical Paper by Hurley (2008). This modelling system has already been extensively applied worldwide, exhibiting good agreement when compared against observational data (Belhout et al., 2018; Duque et al., 2016; Fridell et al., 2014; P Hurley, 2008; Relvas et al., 2017; Wahid et al., 2013; Xia et al., 2015).

  1. Figure 4. The concentration isopleths do not seem to be a good match with the wind roses.  Why is that?  The concentration isopleths seem reasonable given the locations of sources, but it would be useful to see a topographic map (such as in attached file showing Terni and the met site) in the supplemental material in order to better understand the air flows.  Narni, the source of meteorological data, has a rather different alignment to the topography than does Terni.  The authors state that the Narni data were used to understand wind patterns in the area.  I’m not sure that the Narni data will be representative of what is going on in Terni.  The Narni wind rose seems to reflect the ridge of higher elevation running NW to SE there.  That same topography is not seen at Terni.  It would be good to know the precise location of the met station.  The coordinates are only given to the nearest minute, and according to those data it is located on the edge of the Nera River valley that has a N-S orientation as it cuts through the terrain, so channelling of winds is possible.  If the wind rose at Narni is representative one would expect the concentration isopleths to be aligned with higher concentrations to the N and to the SSW, but in fact in Figure 4 they are aligned along an E-W axis.

Regarding this point, there is a misunderstanding. The only available database of wind data for the Terni area is the meteorological station located within the municipality of Narni. The city of Narni does not include only the old town (in the upper side of the hill, as showed by Rev.#2 in the reported map), but also the area named “Narni Scalo” which is included in the above-mentioned valley, and directly borders with the city of Terni. Wind data were acquired through the regional system (https://www.regione.umbria.it/ambiente/servizio-idrografico) and the list of all available meteorological stations can be found at: https://annali.regione.umbria.it/. As you can verify, the meteorological station in the city of Terni (Coordinates WGS84: 42.5597222, 12.6502777) does not include the anemometer among its sensors, while the one located in Narni Scalo, among other data, is providing wind velocity and wind direction data. To better explain this point, the exact coordinates of the meteorological station of Narni Scalo are now reported within the manuscript (Coordinates WGS84: 42.5513888, 12.5441666) and an additional figure in supplementary material shows the exact position of the anemometer used in the study, which is located in the same valley of Terni, at 3 km from its borders, and 6 km from the city center. We are confident that this contextualization and the detailed description of the available database clarify the wind dynamics to the readers.    

  1. Figure 5.  At all four districts, the pattern seems to indicate lower concentrations in summer and higher concentrations in winter.  Figure S1 shows that home heating is a larger percentage of total PM10 emissions in winter and steel process emissions are a larger percentage in summer.  However, this begs the question of the magnitude of the emissions over time.  Figure 1 gives total PM10 emissions, but there is insufficient information to show how the emissions from each source category changes over time.  This seasonality also raises the question of wintertime temperature inversions in an area with significant topography.  The authors should discuss whether such phenomena are known to occur in the area and how that might affect their findings.  They should also address how effectively TAPM may or may not be in capturing potential inversions. At lines 361-362 there is evidence that TAPM may not be capturing this.

Rev.#2 is right, we did not mention the important phenomenon of temperature inversion. This is a very good point to be added in the first part of the discussion, where the seasonality issue is tackled, and some differences are recorded between ECS and TAPM. The following sentence has been included in the discussion: “Furthermore, it is noteworthy how the seasonality is more evident in ECS than TAPM. This condition is probably connected to the temperature inversion, a phenomenon that could easily occur in a significant topography like Terni valley, and which is detected by ECS and not evidenced by TAPM.”

  1. An alternative presentation to Figure S1 is a bar chart showing total emissions by month (if possible) with each bar subdivided by color into the various emissions categories. That would provide a broader perspective on the emissions.  At line 176 the authors state, “Annual data from the ARPA Umbria atmospheric emission inventory … for each pollutant and activity sector, were spatially and temporally disaggregated using a top-down approach to obtain the required resolution for the selected simulation domain.”  More of these data should be shown in the supplemental information.

Figure SI.1 (now converted in SI.2) pertains to the methodological section and it has been produced to present: i) the percentage contribution of each sector in the study area for the considered year, deriving from the available database; ii) the quantitative overview on particles emissions as recorded by local control network. If we consider the available recorded data (annual emission inventory), the only possible representation of such kind of data is the two seasonal scheme we presented. Through the model's preprocessor (TAPM) it would be also possible to transform annual emissions into hourly emissions, using different profiles depending on the emission source. Nevertheless, this analysis cannot be presented in the methodological section because it would not pertain to the starting data framework (materials) but it would be a result (elaboration). Nevertheless, we modified Figure SI.1 (now SI.2) following your suggestion, and we are proposing an alternative representation: instead of the pie chart, a stack bar chart is now shown for the two main seasons. The spatial and temporal disaggregation of data has been described in section 2.2. To better explain the spatial disaggregation the following sentence has been added: “Regarding the spatial disaggregation, all SNAPs included within each cell give its contribution to the relative cell in terms of emissions.”.

  1. Since seasonal differences are seen in the patterns of concentrations, it would be useful to see the wind roses by season. Another helpful piece of information might be a diagnostic wind field model that would incorporate other met stations, including winds aloft, and would give insights into air flows in this complex valley.  That type of information would be very helpful in understanding the results.

We produced a new Figure 1 to describe the main winds directions and velocities for the two main seasons considered in the study: Autumn – Winter (AW); and Spring – Summer (SS). The figure has been modified also according to Rev.#3 suggestions.

  1. The Steel Factor (SF) analysis is an attempt to look at source-receptor relationships. Other options include traditional Gaussian plume modeling, Eulerian grid modeling, or receptor modeling (such as CMB, although it is unclear from Figure S4 whether sufficient speciation data is available).  The authors should justify the advantages the SF approach has in comparison to other approaches.

We evaluated the possibility to implement other kinds of analysis for the source-receptor relationships with the Steel Plant. Nonetheless, we found a fundamental lack of information regarding the stack emissions from the Steel Plant. The local environmental control agency (ARPA), even if should have provided the requested data, when we asked for the detailed information (quantitative emissions at stack levels, and relative fumes temperatures and velocities, from all the different stacks in the huge industrial area) they did not provide it. Thus, through this alternative solution, we could indirectly perform this kind of analysis. We argue this could be an example of an alternative and low-cost way to evaluate the PM dispersion from fixed sources, useful when there are little available data. Furthermore, differently from other techniques, the main aim of SF introduction is the simultaneous comparison of the three datasets shown in the study. The following sentences have been added. In the Methodology section: “The factor has been created to provide the analysis of a source-receptor relationship.”; And in the Discussion: lines 534-540 – “Furthermore, to simultaneously compare the three datasets the SF was introduced and the peculiar discrepancies recorded between ECS and TAPM are confirmed by this further source-receptor relationship analysis. The choice of the use of a newly introduced factor, instead of a Gaussian plume, Eulerian grid, or other receptor modeling has been mainly driven by the lack of information provided by the local environmental control agency (ARPA Umbria). No specific data at stack levels for the steel plant was provided.”

  1. Figure 6. The description of this figure seems a little muddy and rambling, and I’m not sure I fully understand what the authors are trying to say with this figure.  On first impression it seems to me that there is some agreement between the leaf measurements and measured air concentrations, but the relationships with TAPM are less clear.  Am I understanding this correctly?

Rev.#2 is right, the caption was unclear. We reworded it. Your impressions are correct but incomplete due to the inaccurate description of the figure. We are confident that the new description is now clearer and more precise.

  1. I was glad to see that the authors discuss the impacts of rain events and potential wash off. I was thinking of this as I read through the MS and was pleased that was considered.  The authors cite their earlier work on leaf sampling, but it is worth mentioning in the methods section that oaks were sampled, and the time period when the leaves were present on the trees should be specified.

We thank Rev.#2, the important information regarding the sampled species (Quercus ilex, L.) was missing. We added this information in the methodology section and we also gave additional temporal information on gathered leaves (section 2.3).

  1. Using total annual emissions from the steel plant (and other sources) and parsing them uniformly over the year, may be an important source of error. It is unclear how the temporally and spatially disaggregated emission (see comment 8) were used in the modeling analysis.  I don’t know the regulations and reporting requirements in this jurisdiction, but in many places stack by stack emissions over time, including stack release parameters, are considered public information that can be obtained with an inquiry.  If that is possible here, it should be done.  It would also be useful if the release parameters were stated.  Industrial sources typically emit through taller stacks, and the emissions can be dispersed before reaching the ground and vegetation.  On the other hand, traffic and household emissions occur near the ground level (and near the breathing zone and near leaf surfaces).  Since it appears that emissions were apportioned to grid cells (and release parameters not specified), how does the absence of release parameters factor into the analysis and interpretation that has been presented?

As reported for comment 8, the detailed emissions data stack by stack was not provided by the agency responsible for the data collection and dissemination. For this reason, considering the typology of data available we cannot provide further analysis regarding the emissions from the steel plant. As explained in comment 8, the introduction of SF was also justified by the absence (the non-provision) of more detailed data allowing a different/deeper analysis.

Reviewer 3 Report

Dear authors,

you presented topic on urban-industrial environment and compared data of dispersion modeling with tree leaves deposition.

In general, article is good, but it can be improved.

First, study area is correctly described in text. But for broader audience picture of Italy and where this region is located in Italy should be presented. 

Line 132 - explain which GIS analysis. There is many many options available and results could affect your study. 

Lines 133 to lines 140 - please try to represent in table for better visibility of classes and sub-classes. Like this it is realy hard to follow. Please avoid term "around". Scientific paper should have clearly defined percentage range. Therefore, define it correctly. 

Describe classes correctly. What does Nature includes? What does Agriculture includes? Please see Corine Land Cover for better understanding of classes.

Is there some kind of river on your study area? Why not exclude water bodies from Nature class?

Line 175 - which mobile sources are you taking into consideration? How are you putting location on mobile sources (how do you know in which cell grid to put your measurement)?

Why are you using measurements from 2012 and 2013? Is there any new data? (lines 224)

Lines 269 - 276 try to present as map - it is easier to track. Like this readers have no idea which area is Borgo Rivo or Verga (readers outside this region have a hard time to visualize)

Figure 6 is low resolution. Try to enhance (follow the manuscript guidelines)

Discussion is well written.

Support conclusion with main results (numerically). 

Author Response

Rev # 3

Dear authors, you presented topic on urban-industrial environment and compared data of dispersion modeling with tree leaves deposition. In general, article is good, but it can be improved.

Dear Rev.#3, the authors thank you for these considerations.

First, study area is correctly described in text. But for broader audience picture of Italy and where this region is located in Italy should be presented. 

As suggested, also according to Rev.#2 comments, Figure 1 has been modified and improved with new images and information.

Line 132 - explain which GIS analysis. There is many many options available and results could affect your study. 

We used the open software QGIS to perform this analysis. As suggested by Rev.#3 we added the mandatory citation in the text.

Lines 133 to lines 140 - please try to represent in table for better visibility of classes and sub-classes. Like this it is realy hard to follow. Please avoid term "around". Scientific paper should have clearly defined percentage range. Therefore, define it correctly. 

We added a new Table (Table 1) to give better visibility of classes and sub-classes. The text has been modified according to the suggestion. Furthermore, within the table descriptions for Nature and Agriculture classes the Corine Land Cover classification has been adopted.

Describe classes correctly. What does Nature includes? What does Agriculture includes? Please see Corine Land Cover for better understanding of classes.

Please, see the answer to the previous comment.

Is there some kind of river on your study area? Why not exclude water bodies from Nature class?

As described by the new Table 1, the river of the city, namely “Nera” river has been included within the Nature class (“Water courses” Land Cover Corine Class). We decided to group it with the large “Nature” group since we do not see any potential different contribution of the river to PM emissions in the study area.

Line 175 - which mobile sources are you taking into consideration? How are you putting location on mobile sources (how do you know in which cell grid to put your measurement)?

This step has been firstly explained between lines 157-151. As mobile sources we considered the vehicles passing through the main traffic roads. The major roads have been classified into three categories (depending on the number of vehicles per day – see the text), and the total amount of mobile sources from the emission inventory (SNAPs 7 and 8, Table 2) has been allocated according to this classification. See also lines 218-222.  

Why are you using measurements from 2012 and 2013? Is there any new data? (lines 224)

For ECS there are of course newer available data. On the other hand, for Leaf deposited PM the most complete series of data is for 2012 and 2013. We based our study on a previous analysis (see Sgrigna et.al 2015, on Environmental Pollution), integrated by an unpublished dataset from the same period.

Lines 269 - 276 try to present as map - it is easier to track. Like this readers have no idea which area is Borgo Rivo or Verga (readers outside this region have a hard time to visualize)

We do understand the difficulties in the visualization of this step. Nevertheless, Figure 1 has been enhanced and integrated with new information, and it better shows the relative position of each sampling area concerning the Steel plant. Furthermore, the following sentence has been added: “In Figure 1 the relative position of each district respect to the Steel Plant area is shown.”

Figure 6 is low resolution. Try to enhance (follow the manuscript guidelines)

Figure 6 has been enhanced following the manuscript guidelines.

Discussion is well written.

We thank Rev.#3 for this consideration.

Support conclusion with main results (numerically). 

Specific numerical information has been added to the conclusions.

Round 2

Reviewer 2 Report

General Comments

The authors have done a good job of addressing my comments and making revisions accordingly.  I think there are still a few minor tweaks that should be made.

Specific Comments

  1. In their responses, the authors state: “The emissions are provided on an hourly basis by disaggregating temporally the annual emissions using profiles according to the emission activity sector.  Background air pollutants concentrations were also provided to TAPM based on air quality available data for Terni.”  To better understand the results, it would be helpful to know more about how the annual emissions were disaggregated and the time frame(s) for the Terni air quality data.  Perhaps this information could be added to the supplementary material.
  2. It was very helpful for me to see the link to the met stations, including exactly where they are located and what parameters are measured. That information helps the reader better understand the concentration isopleths, and I’m glad to see that a link was provided.
  3. I still have some concern (and I imagine I am not the only one who would notice this) that the wind roses in Figure 1 do not match up well with the isopleths in Figure 4. The predominantly N-S winds shown in the wind rose would not tend to bring the steel plant plume towards the city center.  On the other hand, winds channeled through the river valleys to the east of Terni would bring the steel plant plume towards the city center.  Is TAPM capturing that effect?  It seems plausible given the 1 km resolution in topography (actually the resolution is unclear, see inserted paragraph starting at line 183)   I think the authors should address this apparent contradiction.
  4. Line 185. “The data is” should be “The data are”
  5. I probably should have mentioned this previously. I’m not sure it can be addressed in this paper, but the authors might want to consider it in future analyses.  There are models that will simulate dry (and wet) deposition to surfaces.  Probably the most widely used approach was developed by Wesely and Hicks and is used in AERMOD.  I’m sure the authors are aware of these approaches, but here is a presentation that lays things out nicely (https://gaftp.epa.gov/Air/aqmg/SCRAM/workshops/2018_RSL_Modelers_Workshop/Presentations/2-5_2018_RSL-Particle_Deposition.pdf).  TAPM gives air concentrations, but going a step further, deposition could be modeled and compared with the leaf measurements.  One even further step would be to decrement modeled deposition by the number and intensity of rain events that would contribute to washing particles from leaves.

Author Response

General Comments

The authors have done a good job of addressing my comments and making revisions accordingly. I think there are still a few minor tweaks that should be made.

We thank you for this comment.

Specific Comments

  1. In their responses, the authors state: “The emissions are provided on an hourly basis by disaggregating temporally the annual emissions using profiles according to the emission activity sector. Background air pollutants concentrations were also provided to TAPM based on air quality available data for Terni.” To better understand the results, it would be helpful to know more about how the annual emissions were disaggregated and the time frame(s) for the Terni air quality data. Perhaps this information could be added to the supplementary material.

Thank you very much for this remark.

We realize that in fact we didn’t consider the TAPM optional concentration background file (https://www.cmar.csiro.au/research/tapm/docs/tapm_v4_user_manual.pdf) containing hourly concentrations from an air quality station, this was ratified in the manuscript (lines 193-195). Regarding disaggregating temporally emissions, as suggested we added in the supplementary material a table with the weight attributed to each hour of the day by activity sector (SNAP macro-sector). Please note that no distinction is made for weekdays and weekends. To clarify all this step a large explanation has been added in section 2.2 (lines 213 – 217): “For the emissions temporal disaggregation on daily base, the weight attributed to each hour of the day by activity sector followed the standard profile provided by TAPM. The disaggregation of emission data along the 24h per each SNAP (macro-sector) that has been implemented is showed in Figure SI.2. No distinction is made for weekdays and weekends.”. Furthermore, an additional Figure on Supplementary Information section has been added (Figure SI.2).

  1. It was very helpful for me to see the link to the met stations, including exactly where they are located and what parameters are measured. That information helps the reader better understand the concentration isopleths, and I’m glad to see that a link was provided.

            We thank Rev.#2 for this comment. We are glad to know that the additional text and figure provided can help to better understand all the work.

  1. I still have some concern (and I imagine I am not the only one who would notice this) that the wind roses in Figure 1 do not match up well with the isopleths in Figure 4. The predominantly N-S winds shown in the wind rose would not tend to bring the steel plant plume towards the city center. On the other hand, winds channeled through the river valleys to the east of Terni would bring the steel plant plume towards the city center. Is TAPM capturing that effect? It seems plausible given the 1 km resolution in topography (actually the resolution is unclear, see inserted paragraph starting at line 183) I think the authors should address this apparent contradiction.

            We agree with Rev.#2 regarding this apparent mismatch between the wind roses and the representation of modelled particles dispersion. We define it as “apparent” because it looks like this if we just compare two of the images showed in the paper (Figures 1 and 4). Both images were realized by considering a large temporal set of data, they cannot provide a punctual information but just a rough idea of how the main winds are blowing (Figure 1) and how the model shows its output (Figure 4). The real comparison between ground and model data is given by Figure 5, where ECS and ARPA data are directly related. TAPM is probably capturing the wind-channeling effect through the eastern valley of Terni (by computing in its elaboration all wind data provided by us) nevertheless, we cannot probably appreciate this aspect just by observing the two season-averages wind roses and model dispersion representation (Figures 1 and 4). We argue that it would be possible through a more local-based and finer analysis, specifically focused on the south-eastern side of the study area.

  1. Line 185. “The data is” should be “The data are”

            Changed in the text.

  1. I probably should have mentioned this previously. I’m not sure it can be addressed in this paper, but the authors might want to consider it in future analyses. There are models that will simulate dry (and wet) deposition to surfaces. Probably the most widely used approach was developed by Wesely and Hicks and is used in AERMOD. I’m sure the authors are aware of these approaches, but here is a presentation that lays things out nicely

(https://gaftp.epa.gov/Air/aqmg/SCRAM/workshops/2018_RSL_Modelers_Workshop/Presentations/2-5_2018_RSL-Particle_Deposition.pdf). TAPM gives air concentrations, but going a step further, deposition could be modeled and compared with the leaf measurements. One even further step would be to decrement modeled deposition by the number and intensity of rain events that would contribute to washing particles from leaves.

            We thank Rev.#2 for this valuable suggestion. We are aware about tools like AERMOD which can also calculate / infer the wet and dry deposition to surfaces. Unfortunately, we cannot address this topic in the present research, but we already considered to use this approach in future studies that our research groups are planning to develop. Nevertheless, we added the following sentence in the Conclusions section: “A future perspective of this research is the further comparison of our results with a dispersion model able to evaluate also the dry and wet deposition at ground level, such as e.g. AERMOD (American Meteorological Society/ Environmental Protection Agency Regulatory Model Improvement Committee Dispersion Model) can provide.”.

Reviewer 3 Report

Dear authors,

you did a great job on correcting manuscript. You have answered all my considerations.

This manuscript can be accepted.

Best regards

Author Response

Dear authors,

you did a great job on correcting manuscript. You have answered all my considerations.

This manuscript can be accepted.

Best regards

The authors thank Rev.#3 for this last consideration.

Our best regards.